# Is Continuous ECG Recording on Heart Rate Monitors the Most Expected Function by Endurance Athletes, Coaches, and Doctors?

**DOI:** 10.3390/diagnostics10110867

**Published:** 2020-10-23

**Authors:** Robert Gajda

**Affiliations:** Center for Sports Cardiology at the Gajda-Med Medical Center in Pułtusk, ul. Piotra Skargi 23/29, 06-100 Pułtusk, Poland; gajda@gajdamed.pl; Tel.: +48-604286030; Fax: +48-23-6920199

**Keywords:** heart rate monitor, ECG, portable/wearable monitoring system, endurance running, cycling, triathlon, long-term assessment, arrhythmia, exertion rhythm disorders, QARDIO MD system

## Abstract

Heart rate monitors (HRMs) are important for measuring heart rate, which can be used as a training parameter for healthy athletes. They indicate stress-related heart rhythm disturbances—recognized as an unexpected increase in heart rate (HR)—which can be life-threatening. Most HRMs confuse arrhythmias with artifacts. This study aimed to assess the usefulness of electrocardiogram (ECG) recordings from sport HRMs for endurance athletes, coaches, and physicians, compared with other basic and hypothetical functions. We conducted three surveys among endurance athletes (76 runners, 14 cyclists, and 10 triathletes), 10 coaches, and 10 sports doctors to obtain information on how important ECG recordings are and what HRM functions should be improved to meet their expectations in the future. The respondents were asked questions regarding use and hypothetical functions, as well as their preference for HRM type (optical/strap). Athletes reported distance, pace, instant HR, and oxygen threshold as being the four most important functions. ECG recording ranked eighth and ninth for momentary and continuous recording, respectively. Coaches placed more importance on ECG recording. Doctors ranked ECG recording the highest. All participants preferred optical HRMs to strap HRMs. Research on the improvement and implementation of HRM functions showed slightly different preferences for athletes compared with coaches and doctors. In cases where arrhythmia was suspected, the value of the HRM’s ability to record ECGs during training by athletes and coaches increased. For doctors, this is the most desirable feature in any situation. Considering the expectations of all groups, continuous ECG recording during training will significantly improve the safety of athletes.

## 1. Introduction

Heart rate (HR) monitoring during training in endurance sports is a standard method for controlling intensity. It was introduced into training long before heart rate monitors (HRMs). Originally, athletes used a sweep-handed watch and, directly after stopping, the pulse was measured—usually on the radial artery—to assess the intensity range of their training [1]. The appearance of strap HRMs was a revolution. Additional functions related to global positioning systems (GPSs) give information on the length of the route and speed of the athlete’s movement, along with many other parameters, such as energy expenditure during training [2]. Originally less perfect and burdened with artifacts, HRMs have shown progressively sophisticated technology related to their used materials, and with it, the precision of the recorded parameters has increased [3,4]. In addition, functions such as determining altitude above sea level, water resistance, and GPS-enabled ease of training in all conditions and scenarios have been added [5]. The ability to measure HR in water was another great step, enabling swimmers and triathletes to monitor their training [6].

Heart rate variability (HRV) is a function that allows indirect evaluation of the cause of arrhythmia indicated by an HRM, thus determining the arrhythmia type [7]. In practice, assessing the distance of the R-R points in an electrocardiogram (ECG)—as indicated by the HRV function—does not allow the cause of the rhythm variability to be determined. The inability to distinguish between supraventricular arrhythmias, ventricular arrhythmias, or an ordinary artifact significantly limits its diagnostic value. The possibility for potentially overlooking a life-threatening ventricular arrhythmia is highly likely [8]. The emergence of ECG recording, which is already offered by some HRMs in conjunction with smartphone applications, was a significant advancement. With its ECG application, the Apple Watch Series 4 can generate an ECG recording similar to that of a single-lead ECG. This was a momentous achievement for a wearable device, allowing it to provide critical data for athletes and their doctors; however, the problem of having to stop training to record the ECG remained [9]. Another important function of HRMs, albeit relatively rarely used, is the measurement of HR at rest and at night. Resting HR is an indicator that is used to observe and analyze the athlete’s form—the lower the HR at rest and during sleep, the greater the form [10].

Recently available HRMs can enable continuous ECG recording without training interruption. The QARDIO MD can be described as a typical strap HRM, with the difference being that the information from the transmitter (strap) is transferred to the receiver, which is the Qardio mobile app for the iPhone. After a delay of about 3 min, information from the mobile phone is sent to the “cloud”. Downloading information to the Monitoring Center allows us to not only control the ECG recording—which is continuously recorded—but to automatically recognize life-threatening heart rhythm disorders as well. The Monitoring Center offers an ECG recording of three limb leads (modified leads I, II, III) with automated arrhythmia detection, QRS morphology analysis, P-wave detection (for enhanced automated AF detection), and the possibility of manually assessing the PQ, QT, and ST segments. This provides full control of the ECG recording with simultaneous medical supervision, i.e., online data transmission, but the software is currently only available for physicians and hospitals [11].

As HRMs were developed for healthy athletes, the question of their use in ECG recording has become pertinent. The Holter ECG is used to assess arrhythmias and recognizes both the type of arrhythmia and its location with high probability [12]. Many athletes use HRMs daily and have observed unexpected increases in HR during training, suggesting the presence of an arrhythmia, causing them to undergo extensive and unnecessary diagnostic testing, including electrophysiological tests. Ninety-nine percent of anomalies in HR are due to technical problems (artifacts) that mimic an arrhythmia [13]. Therefore, further investigation into the value of ECG monitoring within HRMs for athletes, coaches, and doctors is required.

The aim of this study is to investigate the opinions on the development of HRMs amongst endurance athletes, coaches, and doctors to determine whether the ECG recording function is considered important.

## 2. Materials and Methods

### 2.1. Group Characteristics

We conducted three surveys among 100 endurance athletes aged 21–57 years (35.5 ± 4.5 years) who were daily users of sport HRMs and were under the care of our Sports Medicine Clinic. The study group included 76 long-distance runners (50 males, 26 females), 14 cyclists (11 males, 3 females), and 10 triathletes (9 males, 1 female). Most of the athletes were under long-term observation—for up to 10 years—and had participated in previous studies related to the use of HRMs, including their usefulness in the assessment of arrhythmias or exercise intensity [14,15,16]. The same surveys were conducted amongst 10 coaches aged 26–60 years (47.0 ± 7.5 years), and 10 doctors (33–60 years, 52.0 ± 7.0 years) who were training and examining endurance sportsmen on a daily basis.

Questionnaire One contained 11 questions concerning the usefulness of individual functions, even hypothetical ones, possessed by modern HRMs in a typical situation and the hypothetical assumption of suspicion of arrhythmia in an athlete. The interviewers, assessing the importance of the functions possessed by HRMs, assigned them an importance ranking from 1–11, where 1 point (p.) meant the highest and 11 meant the least important function. The questions concerned functions such as (a) distance; (b) speed/pace; (c) current HR; (d) average training HR; (e) number of calories consumed during training (active kcal); (f) recording of the current ECG “on demand”; (g) continuous ECG recording; (h) the moment of reaching anaerobic threshold (AT) (lactate threshold); (i) altitude (meters above sea level (MASL)); (j) HRV; and (k) 24-h HR measurement.

Athlete inclusion criteria were the use of HRMs, regardless of the brand, for a minimum of 2 years and minimal personal experience with strap and optical HRMs. Some athletes had been using HRMs for more than 10 years. The second questionnaire assessed HRM preferences—optical (OHRM) versus strap (SHRM)—of the athletes, coaches, and doctors in everyday training versus training with the hypothetical assumption of suspicion of heart rhythm disturbances in the athlete (Figure 1). Both types of HRMs are assumed to be valid and resistant to artifacts. Such an assumption was adopted due to common concerns among respondents about artifacts that distort HR values, to a greater extent in OHRMs, and are familiar to their users [13].

Knowing the results of the preferences in the use of HRMs, all surveyed groups in Questionnaire Three were asked, in detail, about the reasons for their preferences (OHRM versus SHRM selection).

### 2.2. Statistical Analysis

Normal distributions were analyzed using the Shapiro–Wilk test. As age and experience—both with OHRMs (years) and SHRMs (years)—were characterized by the lack of a normal distribution, descriptive statistics were assessed, namely the median and quarter deviation. Correlations between ranks of HRM functions were measured by Spearman’s rank correlation coefficient. Statistical significance in difference in OHRM/SHRM preference depending on the health status of an athlete (healthy vs. suspicion of arrhythmia) was established using chi-squared tests. The average rank of the HRM individual function for every group was set using the mean value. All statistical calculations were performed using STATISTICA 12 (StatSoft, Krakow, Poland). The significance level was set at *p* < 0.05.

### 2.3. Ethical Approval

This study was approved by the ethical review board of the Bioethics Committee of the Healthy Lifestyle Foundation in Pułtusk (EC 6/2020/medicine/sports, approval date: 1 July 2020). All experiments and procedures were conducted in accordance with the Declaration of Helsinki. The athletes provided their written informed consent prior to participation in the analysis and gave permission for their data to be published.

## 3. Results

Analyzed answers to Questionnaire One can be found in Figure 2. The data analysis showed that the groups—athletes, coaches, and doctors—had slightly different expectations regarding the importance of the possessed and hypothetical functions and, moreover, the preferred direction of HRM development. There was a strong positive correlation between the ranks of athletes and coaches (r = 0.93); a low negative correlation between the ranks of doctors and athletes (r = −0.27); and an insignificant negative—or even a lack of—correlation between the ranks of coaches and doctors (r = −0.13).

For athletes, the most important functions were the accuracy of the measurements regarding distance, speed/pace, and current HR, and indication of the attainment of anaerobic threshold (first to fourth places, respectively); ECG recording ranked eighth and ninth for on-demand and continuous recording, respectively. Coaches selected the same top four functions as athletes, only differing in terms of the importance of ECG recording (seventh and eighth, respectively). Doctors’ assessment of the usefulness of ECG was completely different, placing it in positions 1 (continuous recording) and 2 (“on-demand”). The 24-h HR measurement capability and the HRV function ranked third and fourth, respectively.

The same questions, asked in the case of a hypothetical risk of cardiac arrhythmia, had different levels of relevance, especially among athletes and coaches. For athletes, the fourth most important function was continuous ECG recording, with the first three places remaining unchanged. For coaches, ECG recording (continuous and “on-demand”, respectively) was promoted to third and fourth place. Doctors invariably rated the functions describing the work of the heart highly (ECGs, 24-h HR measurement, and HRV). All of the compared groups showed a positive correlation between the given ranks (strong among athletes and coaches, medium among other groups: athletes/coaches, r = 0.84; athletes/doctors, r = 0.55; doctors/coaches, r = 0.60).

The second survey concerned the preferences for the use of HRMs, in terms of OHRM versus SHRM, by athletes, coaches, and doctors in a typical situation and under the hypothetical assumption of suspicion of a cardiac arrhythmia in the athlete (Figure 1). It was hypothetically assumed that both types of HRMs were 100% resistant to artifacts and always correctly indicated assessed parameters. In everyday use, athletes, coaches, and doctors all favored OHRMs (62%, 60%, and 60%, respectively). In the hypothetical heart rhythm disorder situation, the preference of all groups increased in favor of OHRMs (84%, 90%, and 100%, respectively). Observed differences were statistically significant (*p* < 0.001).

Questionnaire Three asked the participants about the reason for their HRM preference (OHRM versus SHRM), assuming that both HRM types had the same functions and level of artifact resistance. The collective results are presented in Table 1.

The survey showed that the two main reasons for selecting optical HRM are related to the all-day comfort of use. Habit and confidence in the indicators were the main reasons for choosing the strap HRM. Most participants (5:2) stated that they would prefer to use an optical HRM in the future. The characteristics of the group and the detailed answers of each respondent to most of the questions asked are found in Appendix A.

## 4. Discussion

### 4.1. Analysis of Results

The analysis of information obtained from 120 people (athletes, coaches, and sports doctors) with many years of experience in personal use of HRMs showed different expectations regarding the direction of development for the functions of modern HRMs. Their participation in previous HRM studies on the differentiation of arrhythmias with artifacts was not without significance when answering the questionnaires. The potential health condition of an athlete using an HRM had an impact on the assessment and usefulness of the individual functions of HRMs. While, for athletes, the most important function was to assess the distance, speed, accuracy, and HR during training, the inclusion of potential heart disease with accompanying cardiac arrhythmias “shifted” the continuous ECG recording function quite clearly in the hierarchy of importance (from 9 to 4). This approach seems perfectly justified. Athletes put their training first. Being ‘healthy’, they do not treat HRMs as medical devices that serve to protect their health. For coaches, the important elements of HRMs were shown to be speed and accuracy in measuring the route, HR during training, and the possibility of determining the oxygen threshold. Regardless of the athlete’s health, coaches only appreciated the possibility of continuous ECG recording by HRMs to a slightly higher extent than athletes. This can also be understood by the assumption that trained healthy athletes aim to achieve sports results and are not interested in permanent cardiological control. Doctors, regardless of whether they were dealing with healthy athletes or those suspected of heart rhythm disturbances, ranked the possibility of continuous ECG recording in first place. This is explained by the fact that this is a professional group associated with the training process for whom the athletes’ health is more important than results. All three groups stated that they would prefer to use OHRMs (versus SHRMs), provided that they are reliable (resistance to artifacts); however, this is still an area of difficulty. The indicated reason for such a choice was, among others, the ease of use of OHRMs, both in training and in everyday life (Table 1).

### 4.2. History of Pulse Control from “Fingers on the Radial Artery” to Advanced ECG Recording Technologies

Currently, no HRM in the world has all the functions that respondents were asked about. The indication of oxygen threshold attainment during training by HRMs is a purely theoretical and hypothetical function that is no less desired by athletes and coaches.

The first reports of commercial medical devices for measuring HR came at the beginning of the 18th century [17]. Partially reliable HR control during training appeared with the widespread introduction of sweep hand watches more than two hundred years ago. The athlete had to stop and, most often, count their pulse on the radial artery for ten seconds and then multiply this number by 6 to determine their HR. In this way, they obtained their HR value at the peak of exercise, allowing them to determine the load in the last phase. There was no opportunity to determine the average HR during training; thus, exercise intensity could not be evaluated as a whole.

For doctors, observing the pulse on the radial artery was a factor in making diagnoses long before the advent of classic watches and had nothing to do with competitive sports [18]. Skilled physicians could identify potential arrhythmias and even determine their speed. All HRMs today record HR; however, this is not enough to establish a complete diagnosis of the origin of the rhythm and potential threats to the life and health of the athlete when pathological. There is no ability to determine whether an arrhythmia at a given time is caused by numerous harmless supraventricular beats—or atrial fibrillation—or whether it is a life-threatening ventricular tachycardia [19].

Commonly used SHRMs, which have been commercially available for many years, indicate the correct HR value; however, in the event of an arrhythmia, they are still not a reliable source of information about its type. The introduction of HRV assessment to HRMs has allowed the rhythm “regularity” to be determined; still, it does not define whether a regular or complete arrhythmia is the result of supraventricular/ventricular beats or ordinary artifacts [20]. SHRMs assess the main electric field produced during ventricle contraction. Therefore, they estimate the distance of the R-R points without identifying either P-wave morphology or the QRS complex [21]. This function is completely useless in the case of *commotio cordis*, the mortality rate of which—regardless of the type of HRM or the device controlling the work of the heart (except for the cardioverter-defibrillator)—is very high. However, healthy athletes do not have access to cardioverter-defibrillators [22].

OHRMs have been on the market for about 10 years. The principle of their operation is common, and the accuracy of their measurement is similar to that of the chest SHRM. Optical pulse monitors operate under a completely different principle than SHRMs. While SHRMs work similarly to ECGs, OHRMs use a phenomenon called photoplethysmography (PPG), which constitutes shining light through the skin and measuring the amount of light that is scattered by blood flow. PPG sensors are based on the fact that light entering the body will scatter in a predictable manner as the blood flow dynamics change, such as with changes in the blood pulse rate (HR) or with changes in blood volume (cardiac output). In practice, the optical HR sensor located on the underside of the watch illuminates the blood vessels in the wrist tissue using LEDs, measuring the amount of light dispersed by the blood flow. The advantage of a wrist pulse measurement is convenience, i.e., the ability to measure HR without having to wear a separate strap or other sensors to measure the pulse. Such a watch must be placed directly on the skin with no material in between; occasionally, the watch must be worn higher on the wrist than a normal wristwatch. The sensor detects blood flow through the blood vessels; therefore, the tissue thickness determines the measurement accuracy [14]. OHRMS, as their primary function, can only determine rhythm regularity and, thus, can indirectly be used to make diagnoses, e.g., complete arrhythmia—suspicion of atrial fibrillation [23].

The use of smartphones for arrhythmia monitoring is another advancement for ECG utilization and arrhythmia detection, effectively making the technology available to any smartphone user. Smart wearable technology, while very common, is mostly limited to activity tracking and exercise motivation. Rhythm-strip-generating smartphone products, such as Kardia Mobile by AliveCor and ECG Check by Cardiac Designs, can more accurately detect arrhythmias than wearable monitors. These products, which have been studied in a variety of situations, rely on the use of an external device with metal sensors to create a rhythm strip, which is usually Lead I. A different subset of smartphone products utilize PPG through a phone camera and light to detect atrial fibrillation. Together, these products have created a paradigm shift in rhythm detection and monitoring [9,24].

New electrodes built into the back crystal and digital crown on the Apple Watch Series 4 work together with the ECG app to enable customers to produce an ECG recording similar to a single-lead reading (Figure 3). To take an ECG recording at any time, or following an irregular rhythm notification, users launch the new ECG app on Apple Watch Series 4 and hold their finger on the digital crown. As the user touches the digital crown, the circuit is complete and electrical signals across the heart are measured. After 30 s, the heart rhythm is classified as either AFib, sinus rhythm, or inconclusive. All recordings, their associated classifications, and any noted symptoms are stored securely in the Health application of the iPhone. Users can share a PDF of their results with physicians. Although, similar to the Apple Watch, it is only a record of one limb lead, and it can clearly recognize both the P wave and the QRS complex. This fully corresponds to the classic single Lead 1 ECG recording (Figure 3). The biggest disadvantage of this function is that activity must be paused for recording, contradicting the idea of measurement during training [25].

However, technological advancements brought new solutions including HRMs with applications enabling constant ECG recording during training to the market (Figure 4). The QARDIO MD system (namely, QardioCore ECG with QardioMD remote monitoring cloud based portal) can be described as a typical strap HRM with the difference that the information from the transmitter (strap) is transferred to the Qardio mobile app on the iPhone, i.e., the receiver. After a delay of about 3 min, information from the iPhone is transmitted to the “cloud”. The downloading of information to the Monitoring Center (Hospital, Clinic with QardioMD remote monitoring cloud-based portal) allows the control of ECG, which is continuously recorded, and automatic recognition of life-threatening heart rhythm disorders. The inconvenience of carrying a phone during training is a minor difficulty compared to the enormous amount of information stored, which is transferred online to the “cloud” with a slight delay. The Monitoring Center offers an ECG recording with three limb leads (modified leads I, II, III) with automated arrhythmia detection, QRS morphology analysis, P-wave detection (for enhanced automated AF detection), and the possibility of manually assessing PQ, QT, and ST segments. It is a matter of time until automatic diagnosis of stress ischemia with the QardioMD system will become available. Preliminary studies have shown that it is a system with comparable diagnostic value to the standard 3 Lead Holter ECG monitor [26].

### 4.3. Strap HRMs or Optical HRM?

The surveyed athletes, coaches, and physicians answered this question unequivocally (Figure 1, Table 1). OHRMs, provided that its indications are reliable, are preferred. Wearing a chest strap is troublesome for athletes for numerous reasons, ranging from battery depletion artifacts, interference in the transmission between the strap and the receiver, to the most important for ultramarathon runners: chafing of the skin during long hours of running by a moving strap [14,15]. It is also common to simply forget to put it on during training, which significantly changes the subsequent evaluation of training. Therefore, OHRMs are preferred on the condition that the accuracy of their indications, which remains a problem, is improved [27]. In the past, an issue was the inability to measure HR by HRMs in the water, which was a significant limitation for triathletes and swimmers; however, this problem has now been resolved [28]. OHRMs usually also have a longer battery life which, in 24- or 48-h ultramarathons, is of great importance [29].

### 4.4. HRMs Instead of the Holter ECG?

Sports HRMs were introduced to monitor HR values in healthy athletes and were not meant to be, or compete with, medical devices [30]; however, it is impossible to run daily with an ECG Holter to verify periodic indications of incorrect values while training with a HRM. An algorithm has been developed to deal with such cases [14]. Nevertheless, HRMs should be considered as devices with useful and reliable medical functions, such as reliable ECG recording, intended for use by athletes. Today’s ECGs recorded by HRMs are single limb lead recordings (Apple Watch) or, as in the case of the QARDIO MD system, a 3-limb lead recording (Figure 3A,B). However, this is an evolutionary advancement, introducing devices “for measuring HR for healthy athletes” as advanced medical diagnostic tools for use in sports cardiology [9].

The trouble-free use of HRMs in everyday life makes them a candidate for use as professional equipment that requires special handling skills and professional knowledge for results interpretation (e.g., Holter ECG). It seems that it will only be a matter of time before HRMs will be able to record a 12-lead ECG with the possibility of assessing any ECG features, including the ST segment, which will be extremely important for the diagnosis of exercise ischemia in a classic exercise test [31]. Other data, such as measuring the QT interval or identifying the origin of ventricular beats, will become automatic information related to these recordings.

Anyone, including potentially healthy top athletes, may experience life-threatening exercise arrhythmias [32]. The registration and early interpretation by the HRMs used today by millions of active people may save lives in the future.

It seems that, in the future, the increasingly perfect ECG data recorded on a typical sports HRM will lead to these devices being treated as medical devices necessary for safe, highly professional, and recreational training. The usefulness of these devices in cardiac rehabilitation is undisputed [33]. Currently, we are starting a long-term observation study of patients with Long QT syndrome type VII, employing modern HRMs for use in ultramarathoners (long “battery life”) [34,35].

### 4.5. Bradyarrhythmia on HRMs—A Lot to Show off in Terms of Observing Athletes

Tachyarrhythmias are mentioned constantly regarding the usefulness of HRMs in the assessment of cardiac arrhythmias; however, the wearing of HRMs—as in the case of OHRMs—may contribute to the registration of not only fast rhythms during training but also night bradyarrhythmias, which are common rhythm disturbances in athletes of endurance disciplines [36]. Undoubtedly, this is a space where HRMs, which are used by many athletes, can contribute both to the diagnosis of arrhythmias—if data are “recorded continuously”—and data collection. All of today’s HRM models register a decrease in HR, but they do not all recognize the mechanism by which this decrease occurs (either a conduction block or ordinary bradycardia). In asymptomatic and apparently healthy athletes, either at rest or during sleep, even 15 s pauses in the Holter ECG examination are common. Northcote et al. examined twenty male veteran endurance runners who underwent resting, exercise, and ambulatory electrocardiography testing. Six athletes had a first degree heart block, four had a Mobitz II second degree block, and three had a complete heart block [37].

The “athlete’s heart” and its accompanying bradycardia, or the second-degree A-V block, are physiological adaptations to exercise [38]; however, a break of a few seconds is certainly a pathology that has the potential to be increasingly recognized by athletes using HRMs both in training and at rest and/or sleep. Comfort is also the reason why OHRMs seem to be a more common direction of development.

### 4.6. Other Expectations from HRMs

The indication of the HR value to diagnose oxygen threshold is nowadays information obtained during the ergospirometric examination [39]. There is an enormous demand for this information by athletes and their coaches and, at the same time, a need for a less complicated measuring method. This function ranked 4th among the surveyed athletes and coaches. There is a need—and it cannot be ruled out—that there will also be a determining method in the future.

### 4.7. Strength, Limitations, and Perspectives

This study had two main limitations: first, the questionnaire was relatively modest and included few questions and, second, it was conducted on a small sample size.

This study’s main strengths were, first, that the group commented on the usefulness of HRMs; second, that most athletes had been under the care of a sports medicine clinic for 5–8 years, constantly using HRMS during their training, and thus had vast experience with different HRMs and could recognize their strengths and weaknesses; and, third, that it included coaches and doctors who were personally acquainted with the athletes and cooperated with the Sports Cardiology Center in which these studies were conducted, rendering them up to date with modern HRM technology.

This study’s perspectives included the improvement of the accuracy of already-existing indications in the HRM market and the development of new technology that will allow the widespread use of OHRMs with the function of 24-h ECG recording. Moreover, we are interested in other functions that are not yet available today, such as the expected oxygen threshold indicator. Certainly, there will be new common solutions other than the ones currently available, allowing not only trouble-free ECG recording during training, but also the ability to inform the athlete, coach, and doctor through online means regarding potential threats in the form of heart rhythm disturbances and the emerging features of stress ischemia as well.

The indirect aim of the article was twofold: first, we aimed to investigate and increase the awareness of athletes regarding the need to protect their health during training by controlling heart rhythm and not just heart rate (i.e., ECG recording), and second, we aimed to increase knowledge in this area to protect the lives and health of athletes who sometimes experience tragic cardiac arrhythmias triggered by exercise by encouraging the widespread use of HRM with continuous ECG recording.

## 5. Conclusions

The conducted analysis indicates the diversity of the expectations of athletes, coaches, and doctors as to the direction of the development of modern HRMs. In the case of suspected heart rhythm disorders, the possibility of ECG recording is a priority feature for sports doctors. Considering all expectations, the paradigm will shift to include continuous ECG recording, especially during training. It seems that users prefer OHRMs over SHRMs as they are more comfortable for use in endurance competitions as well as for non-training use.

## Figures and Tables

**Figure 1 diagnostics-10-00867-f001:**
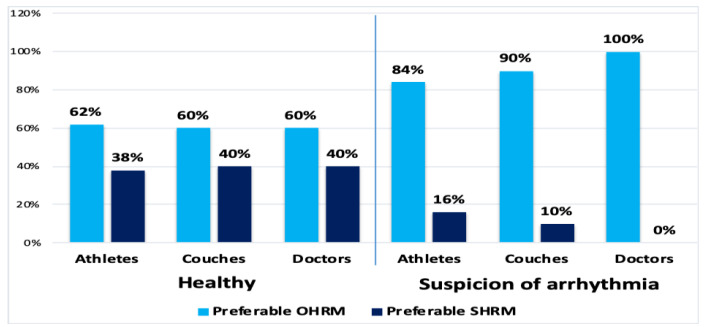
Preferences for the use of HRMs (optical/strap) by athletes, coaches, and doctors in a typical situation and under the hypothetical assumption of suspicion of an arrhythmia in an athlete. Equal and full resistance to artifacts was assumed. Comparison of results of healthy individuals with those of participants with suspected arrhythmia. OHRM, optical heart rate monitor; SHRM, strap heart rate monitor, *p* < 0.001 in every compared pair.

**Figure 2 diagnostics-10-00867-f002:**
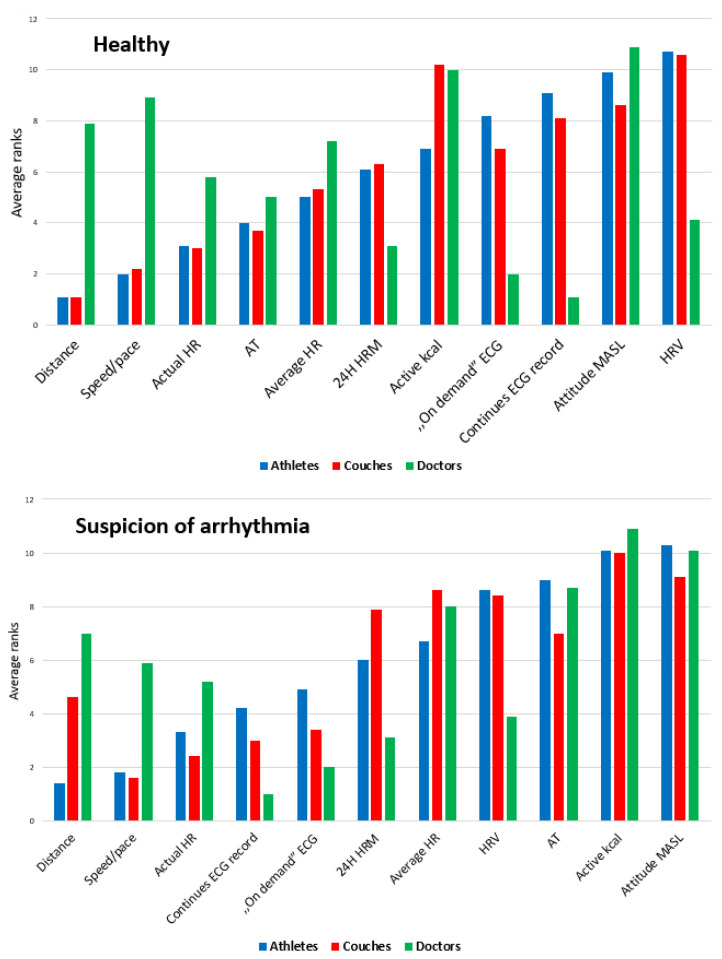
Cumulative results of the survey conducted among athletes, coaches, and doctors regarding the assessment of the importance of individual functions possessed by modern heart rate monitors (HRMs). Two situations are covered: standard use of HRMs and use with the hypothetical assumption of suspected athlete arrhythmia. Data are presented as rankings (mean rankings). Scale rankings 1–11 show decreasing importance of functions. AT: anaerobic threshold; MASL: meters above sea level; HRV: heart rate variability; 24H HRM: 24-h heart rate measurement.

**Figure 3 diagnostics-10-00867-f003:**
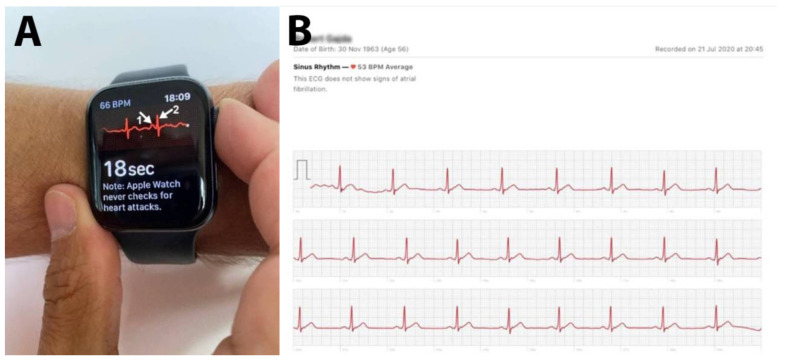
ECG on the Apple Watch, iPhone. (**A**) On the screen, a temporary ECG trace with the morphology of the II limb lead of a classic ECG is shown. Visible: HR 66 bpm. Visible P-waves (arrow 1) and QRS complexes (arrow 2). (**B**) ECG record sent to the iPhone; image taken from the phone screen. Touching the Apple Watch Series 4 Digital Crown completes the circuit and measures electrical signals across the heart.

**Figure 4 diagnostics-10-00867-f004:**
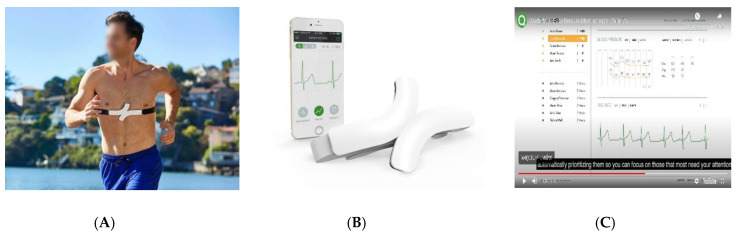
QardioMD ECG solution (QARDIO MD system). (**A**) QardioCore ECG—chest strap with electrode. (**B**) Qardio mobile app (ECG recording on iPhone) and chest strap (QardioCore ECG) with electrode. (**C**) ECG recording on QardioMD remote monitoring web-based portal.

**Table 1 diagnostics-10-00867-t001:** Reasons for preferential use of wrist-worn optical heart rate monitors versus chest strap HRMs by athletes, coaches, and doctors, assuming that the HRMs have the same functions and the same resistance to artifacts.

	Athletes	Coaches	Doctors
Average age (years)	35.5+/−4.5	47.0+/−7.5	52.0+/−7.0
Experience with OHRM (avg. years)	1.3+/−0.5	3.0+/−0.8	2.5+/−1.0
Experience with SHRM (avg. years)	5.3+/−2.0	6.3+/−1.8	5.5+/−1.0
Preferences [OHRM = 1, SHRM = 2]			
Comfort of use during training	1 (88%) *	1 (80%)	1 (80%)
Comfort of use around the clock	1 (95%)	1 (90%)	1 (100%)
Battery life	1 (75%)	1 (60%)	1 (70%)
Skin abrasions from the strap belt	1 (93%)	1 (100%)	1 (100%)
Trend/Fashion	1 (67%)	1 (60%)	1 (60%)
Habit	2 (89%)	2 (90%)	2 (90%)
Confidence in the accuracy of indications	2 (96%)	2 (90%)	2 (90%)
Result: OHRM versus SHRM	5/2	5/2	5/2

* Percentage of votes obtained; 1 = Reason for OHRM preference; 2 = Reason for SHRM preference. OHRM, optical HRM; SHRM, strap HRM.

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
