# Peer review of "Is Continuous ECG Recording on Heart Rate Monitors the Most Expected Function by Endurance Athletes, Coaches, and Doctors?"

_diagnostics, 2020, doi:10.3390/diagnostics10110867_

Round 1
Reviewer 1 Report
This manuscript has been improved a lot. But moderate English changes are required because English language and style is not good neough.
Author Response
Reviewer 1
Comments and Suggestions for Authors
This manuscript has been improved a lot. But moderate English changes are required because English language and style is not good enough.
Response:
Dear Reviewer,
I want to thank you for your thoughtful suggestions and insights. The manuscript has benefited from these insightful suggestions.
The manuscript has been rechecked and the necessary changes have been made in accordance with the suggestions.
I am not a native English speaker, so I entrusted it to editing and linguistic proofreading to a professional English Editing Service - EDITAGE, dealing with proofreading medical texts. This company has high recommendations when it comes to the quality of language correction. I trust that the current form meets your expectations.
I hope you will be satisfied with these explanations.
Kind regards
Robert Gajda
Author

Reviewer 2 Report
The aim of this study was to assess the usefulness of ECG recording functions by sports HRMs among endurance athletes, coaches, and physicians in comparison with other basic and hypothetical functions. For that, the authors conducted surveys and got some interesting results. For athletes, the 4 most important functions were distance traveled, pace, instant heart rate, and information about reaching the oxygen threshold whereas coaches and doctors opined more importance to ECG recording. These results show user preferences but it’s a matter of preference from the perspective of the role. Still, HR monitoring during training in endurance sports is a standard as you mentioned in the article. The results of this study can help strengthen the functions preferred by athletes, but there is very little academic contribution.
Author Response
Reviewer 2
Comments and Suggestions for Authors
The aim of this study was to assess the usefulness of ECG recording functions by sports HRMs among endurance athletes, coaches, and physicians in comparison with other basic and hypothetical functions. For that, the authors conducted surveys and got some interesting results. For athletes, the 4 most important functions were distance traveled, pace, instant heart rate, and information about reaching the oxygen threshold whereas coaches and doctors opined more importance to ECG recording. These results show user preferences but it’s a matter of preference from the perspective of the role. Still, HR monitoring during training in endurance sports is a standard as you mentioned in the article. The results of this study can help strengthen the functions preferred by athletes, but there is very little academic contribution.
Response:
I want to thank you for your thoughtful suggestions and insights. The manuscript has benefited from these insightful suggestions.
The manuscript has been rechecked and the necessary changes have been made in accordance with the suggestions.
The scientific goals (academic contribution ) of the paper are: examine and indirectly increase the awareness of athletes about the need to protect their health during training by control of heart rhythm not only a heart rate. To introduce measures in the future to protect the life and health of athletes who experience tragic arrhythmias induced by exercise by encouraging the widespread use of HRMs with the function of continuous ECG recording. The condition, however, is the universal availability of such sports HRMs, which is a question of the future.
The study is a review of existing devices, but also a great market research, which will allow in the future to implement measures to improve players' awareness of the need to obtain HRM that will have the function of continuous ECG recording. At the moment, there are neither such HRMs nor such awareness among athletes that it is important for their health life. Sports doctors know this very well, which was shown by the study. Their trainers are more aware of this than athletes, which is also shown in the survey. HRMs are also used in the primary and secondary prevention of cardiovascular diseases, which is beyond the scope of this work. It is an example of the use of originally typically sports devices for medical purposes, e.g. cardiac rehabilitation. Description in the text of the article:
…The respondents were asked questions regarding use and hypothetical functions, as well as preference for HRM type (optical/strap). Athletes reported distance, pace, instant HR, and oxygen threshold as the 4 most important functions. ECG recording ranked 8th and 9th for momentary and continuous recording, respectively. Coaches opined more importance to ECG recording. Doctors placed ECG recording as most important. All participants preferred optical HRMs to strap HRMs. Research on the improvement and implementation of HRM functions shows slightly different preferences of athletes compared to coaches and doctors. Suspected arrhythmia increases the value of the HRM’s ability to record ECGs during training by athletes and coaches. For doctors, this is the most desirable feature in any situation. Considering the expectations of all groups continuous ECG recording during training will significantly improve the safety of athletes…
…….. Currently, we are starting a long-term observation of patients with Long QT syndrome type VII, employing modern HRMs used by ultramarathoners (long "battery life")…
I hope you will be satisfied with these explanations.
Kind regards
Robert Gajda
Author

Reviewer 3 Report
The manuscript by Robert Gajda conducted 3 surveys on the importance of ECG recording the ways to access importance of the functions possessed by heart rate monitors for athletes, coaches, and doctors. The results of the surveys were interesting and provided market needs and could be good sources of information for further development of these devices. However, I have the following minor suggestions/questions:
Abstract: ‘Heart Rate Monitors (HRMs) are an ‘ HRM is an …tool
Are figures of apple watch and different devices obtained permission to publish?
Author Response
Reviewer 3
Comments and Suggestions for Authors
The manuscript by Robert Gajda conducted 3 surveys on the importance of ECG recording the ways to access importance of the functions possessed by heart rate monitors for athletes, coaches, and doctors. The results of the surveys were interesting and provided market needs and could be good sources of information for further development of these devices. However, I have the following minor suggestions/questions:
Abstract: ‘Heart Rate Monitors (HRMs) are an ‘ HRM is an …tool.
Are figures of apple watch and different devices obtained permission to publish?
Response:
Dear Reviewer,
I want to thank you for your thoughtful suggestions and insights. The manuscript has benefited from these insightful suggestions.
The manuscript has been rechecked and the necessary changes have been made in accordance with the suggestions.
I am not a native English speaker, so I entrusted it to editing and linguistic proofreading to a professional English Editing Service- EDITAGE, dealing with proofreading medical texts. This company has high recommendations when it comes to the quality of language correction. I trust that the current form meets your expectations.
All figures of the Apple Watch and the other various devices represented in this work have been authorized for publication.
I hope you will be satisfied with these explanations.
Kind regards
Robert Gajda
Author

Round 2
Reviewer 2 Report
The author corrected the manuscript by faithfully reflecting the full reviewer’s comments in a short period, and consequently. The results of this study can help strengthen the functionalities preferred by athletes and improve athlete's awareness that HRMs is important for their healthy life.
Author Response
Dear Reviewer,
Thank you for any comments and time devoted to the analysis of the article.
The article has been reviewed again by a professional editing company cooperating with and recommended by MDPI. I trust that this company has made a professional linguistic correction. I hope that the current form meets your expectations. I enclose a quality certificate
Yours sincerely
Robert Gajda
Author